# Re-Assessing PK/PD Issues for Oral Protein and Peptide Delivery

**DOI:** 10.3390/pharmaceutics13071006

**Published:** 2021-07-02

**Authors:** Randall J. Mrsny, Tahir A. Mahmood

**Affiliations:** 1Department of Pharmacy and Pharmacology, University of Bath, Bath BA2 7AY, UK; 2Applied Molecular Transport Inc., South San Francisco, CA 94080, USA; tahir@appliedmt.com

**Keywords:** oral biopharmaceutical delivery, pharmacokinetics and pharmacodynamics, hepatic portal vasculature

## Abstract

Due to a lack of safe and effective oral delivery strategies for most protein and peptide therapeutics, pharmaceutical drug developers have focused on parenteral routes to administer these agents. Recent advances in delivery technologies have now shown clinical validation for a few of these biopharmaceuticals following oral administration. While these initial opportunities have provided more than just a glimmer of hope within the industry, there are important aspects of oral biopharmaceutical delivery that do not completely align with pharmacokinetic (PK) parameters and pharmacodynamics (PD) outcomes that have been learned from parenteral administrations. This commentary examines some of these issues with the goal of presenting a rationale for re-assessing methods, models, and success criteria to better measure oral protein or peptide delivery outcomes related to PK/PD events.

## 1. Background

For over 100 years, since the first documented description in 1922, extensive efforts have been made to identify safe, effective strategies for the successful oral delivery of insulin [1]. Since that time, more than 850 protein and peptide therapeutics have been approved [2]. Simultaneously, a plethora of approaches to achieve oral delivery of biopharmaceuticals have been described by both academic and industrial scientists using a variety of preclinical models [3]. Markets for oral biopharmaceuticals have been predicted to be worth $43.3 billion USD by 2024 [4]. A PubMed search of the terms “oral protein peptide delivery” and “oral peptide delivery” identifies thousands of entries. Despite the likelihood of many duplications across these searches, the numbers demonstrate the extensive interest in this area. Scanning these publications highlights the wide range of approaches that have been described, including various types of chemical enhancers, cell-penetrating peptides, chelating agents, chemical surfactants, bile salts, mucoadhesive platforms and hydrogels, proteolytic protectants, liposomes, microspheres, and nanoparticles, as well as robotic systems and needle-protruding devices [5,6,7]. While these approaches appear promising in preclinical models, their clinical translation has been complicated by the various barriers to macromolecular uptake presented by the gastrointestinal (GI) tract. Successful translation is further complicated by patient heterogeneity and pathologies, and the challenges of achieving a desired drug biodistribution following oral delivery in humans [8].

With the lack of robust, fully developed oral delivery strategies, pharmaceutical companies logically set out to provide patients with new medicines developed from these biopharmaceuticals using injectable formats, which have been extremely successful. Why, then, is oral biopharmaceutical delivery still of such interest? The answer to this question can be distilled from a variety of factors: benefit(s) to the patient, reduction of environmental burden from needle/syringe use, as well as potential improvements for production, storage, and distribution of oral dosage forms. While intense needle-phobia is a relatively rare condition, patients and their care givers prefer the convenience of an oral dosage form to an injection [9]. Further, oral dosing dramatically reduces the burden of frequent administrations, thereby also having the potential to improve compliance. Oral delivery would also remove the environmental burden of needle, syringe, and autoinjector device disposal [10]. Oral dosage forms of biopharmaceuticals can be prepared using an aseptic process, bypassing the sterile facilities required for injectables. Additionally, oral dosage forms of biopharmaceuticals may not require the same level of cold-chain storage that is needed to ensure stability of injectable formulations. Production costs, in general, should decrease when sterility and reduced cold-chain requirements are relaxed, potentially reducing the overall healthcare costs of oral biopharmaceutical medicines. Thus, the value proposition for oral relative to injectable delivery routes for biopharmaceuticals provides much of the basis for the notion that the oral delivery of protein and peptide therapeutics represents the “Holy Grail” of the pharmaceutical industry [11].

In the absence of oral biopharmaceutical delivery strategies, extensive work was performed to optimize parenteral delivery formats that focused on subcutaneous (SC) injection strategies for self-administration. Molecules were chemically modified to reduce their propensity for degradation and rates of clearance, both of which resulted in decreased injection frequency. Injection devices to simplify administrations were also introduced to increase patient compliance. This combination of approaches successfully expanded the clinical utilization of many biopharmaceuticals, particularly peptides. For example, glucagon-like peptide 1 (GLP-1) has a serum half-life of 1.5–5 min [12]. Modifications made to the GLP-1 molecule, such as lipidation and amino acid exchanges, resulted in dramatically longer serum half-lives that have resulted in commercially successful products, with some reductions in dose-limiting side effects that include nausea, diarrhea, and vomiting [13]. While such chemical modifications led to successful clinical products for parenteral administration, these modified molecules were not necessarily optimized for patient safety and clinical efficacy following oral delivery. Modifications made to these molecules, while improving parenteral administration outcomes, were not necessarily intended to also provide an optimal (or desired) biological action following oral delivery. Thus, it may be time to re-think the characteristics of orally delivered biopharmaceuticals to better emulate their intended physiological function following entry via the GI tract.

## 2. Anatomical, Physiological, and Functional Principles of Oral Drug Absorption

Blood draining from most of the GI tract, beginning in the stomach and ending in the proximal portion of the rectum as well as the gallbladder, pancreas, and spleen, ultimately drains into the portal vein [14]. This anatomical organization establishes the hepatic portal vasculature that supplies the liver with 70–75% of its blood supply, with blood coming from the systemic circulation via the hepatic artery providing the remaining 25–30% [15]. Since the blood volume of the liver accounts for 10–15% of the body’s total ~4.5–5.7 L volume and only 25–30% of that blood enters from the systemic circulation [16,17,18], this makes for highly disparate biodistributions of a biopharmaceutical containing elements of the hepatic portal vasculature following entry into the body from a parenteral administration, such as from an SC injection site, versus oral absorption (Figure 1a). With each passage through the systemic circulation only a fraction of any biopharmaceutical that has been absorbed from a parenteral administration will reach tissues associated with the hepatic portal vasculature that would otherwise receive 100% of that same molecule following its oral uptake [19]. Such differences in this initial distribution are most pronounced for molecules that are sufficiently hydrophilic to be absorbed into blood to enter the hepatic portal vasculature (Figure 1b). The disparity is further compounded by proteins or peptides that act specifically on the liver and/or have a short serum half-life.

Molecules that are sufficiently lipophilic or that have specific interaction(s) with certain immune cell populations would be sequestered into the lacteals and lymphatic vessels of the GI tract. These molecules would bypass the hepatic portal vasculature as they enter the systemic circulation directly at the point where the cisterna chyli/thoracic duct drains into the left subclavian vein [20]. Importantly, portal vein blood flow can deliver an orally absorbed molecule to the liver within minutes while the lymphatic route to the systemic circulation can take several hours [21]. Indeed, this delay in absorption of a lipophilic peptide, cyclosporin, is noted to peak 2.5 h after oral administration [22]. Very large molecules may distribute preferentially to lymphatic relative to blood vessels, but the size for such a discrimination in the intestinal villus has not been described as yet. Based upon these potential differences in fate between hydrophobic versus hydrophilic molecules absorbed in the GI tract, the fate of biopharmaceuticals absorbed into the lymph would more closely match the PK profile from a parenteral administration, such as an SC injection [23]. Thus, if it is critical to mimic the PK profile that is demonstrated by an SC injection, oral delivery that results in lymphatic uptake and not uptake into the hepatic portal vasculature may be a preferable outcome (Figure 1c).

## 3. PK/PD Issues for Orally Delivered Biopharmaceuticals Active in the Gut and Liver

Pharmaceutical companies consider the idea of transitioning an injectable drug into an oral dosage form as a very attractive life-cycle management and product extension strategy for their approved biopharmaceutical products. As stated above, numerous approaches have been described to achieve this outcome, but successes have been rather limited to date. Has the pharmaceutical industry set an unrealistic bar for the success of such efforts? Are we asking too much of oral delivery outcomes with these molecules that were optimized for parenteral administration? Is it even possible to achieve similar outcomes based upon the differences in anatomical, phycological, and functional principles of oral drug absorption outlined above? Using insulin as an example, studies have shown that a permeation enhancing approach could result in 4–5% delivery into the hepatic portal vasculature, but less than half of the absorbed materials were detected in the systemic circulation [24]. Insulin extraction by the liver represents a critical aspect to the receptor-mediated actions of this hormone as it leaves the β-cells in the pancreatic islets of the pancreas to regulate the balance of gluconeogenesis/glycolysis in the liver to control blood sugar. This organization of pancreas–liver communication and insulin actions/regulation would not be recapitulated by insulin entering the body following parenteral administration but would be modeled by entry of this hormone via the hepatic portal vasculature [25]. Considering these points, is it possible that previous data obtained for some oral delivery strategies could have been effective from a PD standpoint, despite what might be perceived as a sub-par PK outcome for the oral delivery of this hormone?

In general, hormones that function in blood sugar regulation, known as incretins, act primarily through receptors distributed in the hepatic portal vasculature. GLP-1, as an example, is produced primarily in the small intestine in response to food ingestion, allowing it to directly target its cognate receptors located in the myenteric plexus neurons in the lamina propria [26]. Indeed, the short half-life of GLP-1 in plasma does not reflect its long-lasting beneficial PD effects [12] as these can perpetuate after a pulsed release and this localized receptor activation scenario. Therefore, one must consider if the modifications that facilitated the development of the currently approved, parenterally administered, biopharmaceuticals have shifted pharmaceutical industry priorities from what is potentially the best way to treat the patient based upon the biology and physiology of a molecule, to what can be effective and safe in the clinic while addressing commercial pressures. Unfortunately, PD/PK outcomes obtained for parenteral administration are often included as a benchmark for assessing the comparative clinical efficacy of the oral products, despite the disparity in the parameters of receptor distribution, duration of action, and vascular biodistribution differences that may exist when compared to a parenteral administration [19]. Successful development of biopharmaceuticals that was primarily achieved through the expertise and talents of chemists and engineers considering parenteral administration issues have taken the pharmaceutical industry to its current state where it successfully treats millions and millions of patients every day. With the advent of promising approaches to now achieve the oral delivery of biopharmaceuticals, however, it may be time to re-think molecule optimization to better mimic the organized nutritional and metabolic responses following focused delivery of these molecules to the intestinal, hepatic, and pancreatic elements of the hepatic portal vasculature.

There is also a potential complication in how orally delivered protein and peptide effectiveness are currently considered. PK/PD associations have been established in preclinical and clinical models specifically directed to ascertain actions following parenteral administration. Such methods were invaluable for the discriminating assessment of the potential candidates required for development. Many of the PK/PD parameters used to assess the parenteral administration of biopharmaceuticals may not be asking the most critical questions, however, when one considers these same molecules for oral delivery. Indeed, under certain circumstances an oral PK profile comparable to parenteral administration may not provide a comparable PD. For instance, systemically delivered anti-tumor necrosis factor (TNF) therapies currently come with a black box warning due to the potential for increased risk of serious infection [27]. In the case of anti-TNF treatment for inflammatory bowel disease (IBD), would there be a safer, equally effective form of this protein class that might be restricted to the intestinal lamina propria following oral delivery as has been suggested [28]? The primary site of action of growth hormone following its release from the pituitary gland involves receptor activation in the liver that results in increased serum levels of insulin-like growth factor I. Thus, oral delivery of growth hormone and its direct uptake into the hepatic portal vasculature could be a more effective way to administer this hormone, shifting the PK/PD relationship from that observed following parenteral administration [29]. In the instances where systemic injection of anti-TNF therapy to treat IBD and growth hormone therapy, oral delivery might provide improved PD with distinct PK from what was identified from parenteral administration. These two examples further suggest that it may be time to re-think targeting and actions within the intestine and associated tissues based upon the therapeutic goal.

## 4. Summary

The premise of this commentary is not to suggest that the outcomes observed for all orally delivered protein and peptides therapeutics will be dramatically different from what has been observed for parenteral administration. For many, maybe most, biopharmaceuticals, transitioning an injectable drug into an oral dosage form could provide a comparable PK/PD outcome to that observed for parenteral administration; such an outcome would theoretically simplify development efforts in the transition from an injectable to an oral dosage form. Rather, our focus is to suggest a re-thinking of PK/PD outcomes for those molecules where the biology of their actions involves or is limited to the intestine and tissues associated with the hepatic portal vasculature. In doing so, it may be appropriate to not only consider the measures by which these assessments are made but also methods that better describe local biological events. In these cases, optimization of both the molecules and the tools to assess the PK/PD relationships that have been established using parenteral administration and the associated systemic outcomes may not align with optimal patient safety and benefit when considering oral delivery. Thus, new thinking on how to measure relevant local PD responses in the absence of a systemic PK that would normally be observed following parenteral administration could benefit the development of the next generation of oral biopharmaceuticals. Further, it may now be possible to explore previously unappreciated PD outcomes with the focused delivery of certain biopharmaceuticals to the intestine and tissues associated with the hepatic portal vasculature, opening new opportunities to address the unmet medical needs that still confront us.

## Figures and Tables

**Figure 1 pharmaceutics-13-01006-f001:**
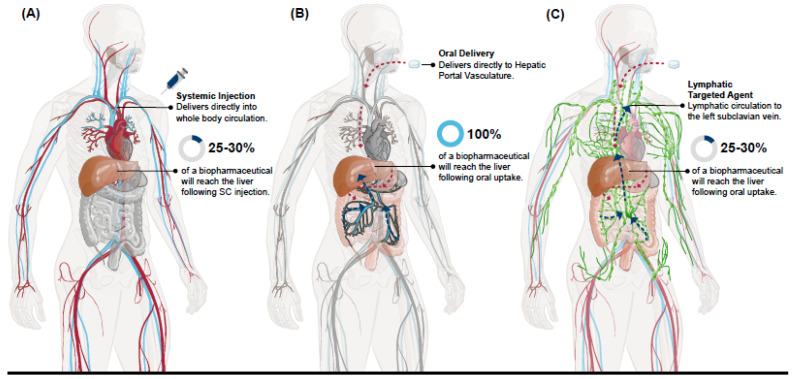
Comparison of parenteral and oral entry routes on initial vascular distribution. Relative distribution of a biopharmaceutical following (**A**) parenteral administration, (**B**) oral uptake of hydrophilic agents into the hepatic portal vein, and (**C**) oral uptake of hydrophobic agents into lymphatic vessels.

## Data Availability

Not applicable.

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
