# Peer review of "Re-Assessing PK/PD Issues for Oral Protein and Peptide Delivery"

_pharmaceutics, 2021, doi:10.3390/pharmaceutics13071006_

Round 1

Reviewer 1 Report

This is a sound commentary that gives some interesting aspects of oral protein delivery and the role of the portal circulation and lymphatics. Whilst I cannot offer any further discussion on this commentary, the authors could perhaps add in some examples of lymphatic drug delivery, if not with proteins, with small molecules (eg doxorubicin delivery). Also, perhaps rephrase the term 'turning a shot' into pill. 

Spelling error line 147.

Author Response

Reviewer 1

The authors could perhaps add in some examples of lymphatic drug delivery

Our response/action(s) We appreciate the reviewer's suggestion. Cyclosporin is a hydrophilic hydrophobic peptide and it's from pharmacokinetic profile nicely demonstrates this delay in systemic uptake. We have added a sentence and reference regarding this point.

….perhaps rephrase the term 'turning a shot' into pill. 

Our response/action(s) We agree that this is jargon and have rewritten the sentence.

Spelling error line 147.

Our response/action(s) Corrected, and we have rewritten the sentence

Reviewer 2 Report

The commentary provides a concise and balanced overview of the challenges in the oral delivery of the protein and peptide drugs. It is well organized and well written. There are some complex and lengthy sentences that can be simplified, to enhance the clarity of the manuscript. In addition, some small typos should be fixed (see below). The contents of Fig. 1 appear to be misleading, and should be corrected:

The values presented in Fig. 1 A-C reflect the first pass only, please state this explicitly. Overall, the liver blood flow accounts for 25% of the cardiac output (and ~25% of the drug in the blood pool will reach the liver with each circulation cycle). The balance between the portal vein vs. hepatic artery perfusion is approximately 75%:25%.

For Fig. 1A, 25% of the drug is expected to reach the liver during the “first pass”.

For Fig. 1B, the size (>16-20 kDa or less than this threshold) and lipophilicity of the drug will govern the effectiveness of its absorption to the lymphatic vs. blood vessels. Thus, much less than 100% of the biopharmaceutical drug would reach the liver during the first pass, especially for the high MW and/or highly lipophilic drugs.

For Fig. 1C, 25% of the drug is expected to reach the liver during the “first pass”, in case of complete absorption via the lymphatic route (then to intestinal lymphatic trunk -> thoracic duct -> systemic blood circulation (at the angle of the left subclavian and internal jugular veins) -> 25% to the liver (portal vein + hepatic artery).

Please correct the statements and the percentages in the Fig. 1A-C.

Lines 110-1 – please fix the typos in this sentence.

Line 128 – “b-islet” please replace with “beta-cells in the pancreatic islets”, or a similar term.

Line 178 – please fix the “in both the case”.

Line 179 – please correct the “oral delivery” typo.

Author Response

Reviewer 2

There are some complex and lengthy sentences that can be simplified, to enhance the clarity of the manuscript.

Our response/action(s) Rewritten sentences have been highlighted.  

The contents of Fig. 1 appear to be misleading, and should be corrected:

The values presented in Fig. 1 A-C reflect the first pass only, please state this explicitly.

Our response/action(s) We have used the term initial vascular distribution as “first pass” hold the connotation of metabolism related to small molecules and this is not the intent of this discussion.   

Overall, the liver blood flow accounts for 25% of the cardiac output (and ~25% of the drug in the blood pool will reach the liver with each circulation cycle).

Our response/action(s) We agree that 25% is a correct number for Fig. 1A and thank he reviewer for catching this mistake. The figure has been corrected.

For Fig. 1B, the size (>16-20 kDa or less than this threshold) and lipophilicity of the drug will govern the effectiveness of its absorption to the lymphatic vs. blood vessels. Thus, much less than 100% of the biopharmaceutical drug would reach the liver during the first pass, especially for the high MW and/or highly lipophilic drugs.

Our response/action(s) We have avoided in assuming anything more than hydrophobic properties to drive a peptide or protein into the lymphatics. We have not been able to find any information regarding the size properties of absorption for the vasculature in the intestinal villus. If the reviewer could provide peer-reviewed data in this area, we will happily include it.  

For Fig. 1C, 25% of the drug is expected to reach the liver.

Our response/action(s) We agree that 25% is a correct number for Fig. 1C and thank he reviewer for catching this mistake.

Some small typos should be fixed: 

Lines 110-1 – please fix the typos in this sentence.

Line 128 – “b-islet” please replace with “beta-cells in the pancreatic islets”, or a similar term.

Line 178 – please fix the “in both the case”.

Line 179 – please correct the “oral delivery” typo.

Our response/action(s) These have been corrected.  

Reviewer 3 Report

Dear Authors,
You have presented an interesting debate in your commentary " Re-assessing PK/PD Issues for Oral Protein and Peptide Delivery". I am sure this will further stimulate the thinking of PK/PD scientist working the area of drug delivery and newer areas of therapeutics like anti-body drug conjugates.
You have sited some references, but due to the lack of case studies in the area the current argument cannot be validated. However, I am sure this will be an interesting area that would see some examples in future possibly from your own venture.  Here is the opportunity to add value to your company and scientific thinking. We look forward to the publications in this direction.
There are several spelling mistakes throughout the manuscript, that need careful reading and correction.
Best wishes

Author Response

Reviewer 3

..several spelling mistakes throughout the manuscript.

Our response/action(s) We believe these have all been corrected.